# Quantifying Crack Self-Healing in Concrete with Superabsorbent Polymers under Varying Temperature and Relative Humidity

**Ahmed R. Suleiman [1], Lei V. Zhang [2] and Moncef L. Nehdi [3],***

1   Golder Associates Ltd., 100 Scotia Court, Whitby, ON L1N 8Y6, Canada; Ahmed_Suleiman@golder.com
2   Department of Civil and Environmental Engineering, Western University, London, ON N6G 1G8, Canada; lzhan666@uwo.ca
3   Department of Civil Engineering, McMaster University, Hamilton, ON L8S 4M6, Canada
*   Correspondence: nehdim@mcmaster.ca

**Abstract:** During their service life, concrete structures are subjected to combined fluctuations of temperature and relative humidity, which can influence their durability and service life performance. Self-healing has in recent years attracted great interest to mitigate the effects of such environmental exposure on concrete structures. Several studies have explored the autogenous crack self-healing in concrete incorporating superabsorbent polymers (SAPs) and exposed to different environments. However, none of the published studies to date has investigated the self-healing in concrete incorporating SAPs under a combined change in temperature and relative humidity. In the present study, the crack width changes due to self-healing of cement mortars incorporating SAPs under a combined change of temperature and relative humidity were investigated and quantified using micro-computed tomography and three-dimensional image analysis. A varying dosage of SAPs expressed as a percentage (0.5%, 1% and 2%) of the cement mass was incorporated in the mortar mixtures. In addition, the influence of other environments such as continuous water submersion and cyclic wetting and drying was studied and quantified. The results of segmentation and quantification analysis of X-ray μCT scans showed that mortar specimens incorporating 1% SAPs and exposed to environments with a combined change in temperature and relative humidity exhibited less self-healing (around 6.58% of healing efficiency). Conversely, when specimens were subjected to cyclic wetting and drying or water submersion, the healing efficiency increased to 19.11% and 26.32%, respectively. It appears that to achieve sustained self-healing of cracks, novel engineered systems that can assure an internal supply of moisture are needed.

**Keywords:** crack; self-healing; environmental exposure; concrete structure; superabsorbent polymer; micro-computed tomography; image analysis

## 1. Introduction

Concrete structures are susceptible to cracking due to various factors such as shrinkage and thermal stresses, mechanical loading, chemical attack, and environmental exposure. The presence of cracks could significantly compromise the durability of concrete structures since they create pathways for the ingress of detrimental agents such as chloride and sulfate ions. Therefore, periodic inspection, preventive maintenance, and repair and rehabilitation are often required. According to the latest Canadian infrastructure report card published in 2019 [1], a concerning portfolio of public infrastructure is in critical condition, requiring immediate rehabilitation. Moreover, most civil infrastructure assets continue to deteriorate prematurely and are expected to fall into similar conditions if appropriate repairs are not made in a timely manner. Similar cases have been reported in other countries. For instance, it is estimated that £40 billion is spent annually in the UK for infrastructure maintenance, of which a substantial portion is used to repair deteriorating concrete structures [2]. On the other hand, according to the ASCE, the US needs to spend $4.59 trillion by 2025 to fix its deteriorating civil infrastructure [3].

In recent years, major efforts have been devoted to developing concrete with self-healing abilities [4–13]. Several studies have investigated the effects of various additives on crack self-healing in concrete under a single environmental condition (typically when concrete is submerged in water). For example, Li et al. [13] studied the effect of ground granulated blast furnace slag on the crack self-healing of cement mortar incorporating crystalline additives under full water submersion. The crack self-healing ability in specimens containing a crystalline admixture was significantly increased compared with that of the control specimens without additives. Similarly, Rong et al. [7] explored the effect of the concentration of Bacillus pasteurii on the crack self-healing of cement-based materials and found that the healing efficiency for wide cracks was much lower than that of thin cracks. Qureshi et al. [6] reported that partially replacing Portland cement with expansive minerals, including bentonite clay, magnesium oxide, and quicklime, improved the self-healing properties of cement paste prisms submerged in water.

Limited studies have so far investigated the efficacy of self-healing in concrete under variable environments. For instance, Roig-Flores et al. [14] studied the self-healing of concrete samples incorporating crystalline additives (CA) exposed to various environments. Their results showed that depending on the exposure condition, the self-healing of concrete incorporating CA could exhibit different healing behavior. Negligible self-healing was found in all samples exposed to constant temperature and relative humidity, demonstrating that the presence of liquid water is essential for self-healing. Sisomphon et al. [15] investigated the effect of environment exposure on the self-healing of cementitious composites with strain hardening ability and made with different cementitious materials. They reported that air curing adversely affected the self-healing process. Likewise, Wang et al. [16] reported no crack healing in bacteria-based concrete maintained at 60% RH and 95% RH. Due to drying shrinkage, the final crack in specimens stored at 60% RH was even wider than their initial size. According to Wang et al. [16], the deficiency of water supply to cracked specimens was the main reason for the lack of self-healing.

The influence of superabsorbent polymers (SAPs) as an admixture in cementitious materials to mitigate shrinkage cracking via providing internal curing has been a subject of research for many years [17–21]. For instance, Jensen and Hansen [18] reported that autogenous shrinkage after the setting of hardening high-performance concrete and subsequent cracking due to restraint could be prevented when SAPs were used as water-entraining admixtures. Similar findings were presented by Soliman and Nehdi [20] and Wang et al. [19]. Superabsorbent polymers are cross-linked and copolymerized polyacrylates that can absorb more than 100 times their own mass of liquid supplied by the surrounding environment and then retain it intact without dissolving [19].

Recently, several studies have investigated the effect of SAPs on autogenous self-healing of cement-based materials [22–26]. For example, Snoeck et al. [22] examined the water penetration in cementitious materials incorporating SAPs using neutron radiography. Their results showed that specimens containing SAPs exhibited lower total moisture uptake than the reference specimens without SAPs. Snoeck and De Belie [23] studied the repeated autogenous self-healing in strain-hardening cementitious materials containing SAPs. They found that SAPs promote autogenous self-healing and mechanical recovery of the strain-hardening cementitious materials. Lee et al. [26] reported that SAPs can re-swell and seal cracks in concrete.

Several studies have also explored the influence of SAPs on autogenous self-healing under different environments such as cycles of wetting and drying, continuous exposure to a constant relative humidity, and/or continuous immersion in water. For example, Lefever et al. [27] evaluated the self-healing of cracks in mortars incorporating SAPs and nano silica under cycles of wetting and drying using microscopy measurements and water permeability testing. Their results showed that self-healing efficiency was markedly increased in mortar specimens containing SAPs. Park and Choi [28] investigated the ability of cementitious materials containing crystalline admixtures and SAPs to attain crack closing due to self-healing when immersed in tap water at a temperature of $20 \pm 3$ °C by means of

water flow test and crack closing test using an optical microscope. They found that using crystalline admixtures with SAPs accelerated the crack sealing.

Sidig et al. [29] investigated the effectiveness of SAPs and superplasticizer in the self-healing of cementitious materials immersed in water at room temperature using X-ray tomography images. They found that self-healing efficiency increased in specimens with 2.2% of SAPs. Snoeck et al. [24] used X-ray computed microtomography to investigate autogenous healing of cementitious materials promoted by SAPs. In their study, three curing conditions under a constant temperature were used, including (a) storing the specimens at a relative humidity (RH) of 60%, (b) at RH > 90%, and (c) exposing the specimens to cycles of wetting and drying (1 hr in water and 23 hr in the lab at 60% RH). Their results showed that specimens exposed to cycles of drying and wetting achieved the highest crack self-healing. Similarly, Snoeck et al. [30] used Nuclear Magnetic Resonance NRA to quantify autogenous self-healing in cement-based materials incorporating SAPs and exposed to similar environments by Snoeck et al. [24]. Chindasiriphan et al. [31] studied the effect of fly ash and SAPs on concrete self-healing ability exposed to either continuous water immersion or wet-dry conditions. They found that specimens exposed to continuous water immersion attained more self-healing than the specimens that were subjected to cycles of wetting and drying.

However, there is a lack of studies in the open literature that have investigated and quantified the change in the entire crack volume due to autogenous self-healing in cement-based materials containing SAPs when exposed to a combined change in temperature and relative humidity. Therefore, in the present study, the entire crack volume changes due to self-healing in cementitious materials incorporating SAPs exposed to an environment with a combined change in temperature and relative humidity was investigated and quantified using X-ray μCT scans. The results of segmentation and quantification analysis were compared with the influence of other environments, including cyclic wetting and drying, and/or continuous water submersion. In addition, the effect of different SAP contents on the mechanical properties and pore structure of concrete was evaluated. The results should provide a robust basis for the quantification of the self-healing potential in cement-based materials incorporating SAPs under combined environmental conditions.

## 2. Experimental Program

### 2.1. Materials

Mortar specimens reinforced with 1% of polyvinyl alcohol (PVA) fibers were made with general use normal Portland cement (CSA A3001). The physical and chemical properties of the cement and sand are summarized in Tables 1 and 2.

**Table 1.** Physical and chemical properties of cement.

| Components/Property | Value |
|---|---|
| Silicon oxide ($SiO_2$) (%) | 19.6 |
| Aluminum oxide ($Al_2O_3$) (%) | 4.8 |
| Ferric oxide ($Fe_2O_3$) (%) | 3.3 |
| Calcium oxide (CaO) (%) | 61.50 |
| Magnesium oxide (MgO) (%) | 3.0 |
| Sulfur trioxide ($SO_3$) (%) | 3.50 |
| Loss on ignition (%) | 1.90 |
| Autoclave expansion (%) | 0.09 |
| Compressive strength 28 days (MPa) | 40.9 |
| Specific gravity | 3.15 |
| Time of setting (min) Vicat Initial | 104 |



**Table 2.** Physical and chemical properties of sand.

| Property | Value |
|---|---|
| Absorption (%) | 1.09 |
| Specific gravity (apparent) (%) | 2.72 |
| Specific gravity (dry) (%) | 2.65 |
| Specific gravity (SSD) (%) | 2.68 |
| Unit weight (kg/m$^3$) | 1512 |
| Materials finer than 75-μm (sieve # 200) (%) | 2.10 |

In the present study, cross-linked sodium polyacrylate SAP (LiquiBlock™ HS Fines) was used. The physical and chemical properties of the SAP are given in Table 3. As shown in Table 4, a varying dosage of SAP expressed as a percentage of the cement mass was incorporated in the mortar mixture. In addition, the amount of water and sand were kept constant at a water-to-cement ratio (w/cm) of 0.35 and sand-to-cement mass ratio (s/c) of 2 in all mixtures to investigate the effect of each SAP dosage on the mechanical properties, pore structure, self-healing capacity of the tested mortars.

**Table 3.** Properties of SAP used.

| Property | Value |
|---|---|
| Particle size distribution (μ) | 1–140 |
| Absorption deionized water (g/g) | >180 |
| Absorption 0.9% NaCl (g/g) | 50 |
| Apparent bulk density (g/l) | ≃540 |
| pH value | 6–7 |

**Table 4.** Mixture design of mortars by mass ratio.

| Mixture | Description | OPC | SAP | Sand | Water |
|---|---|---|---|---|---|
| 1 | 0% SAP | 100 | - | 200 | 35 |
| 2 | 0.5 SAP | 99.50 | 0.5 | 200 | 35 |
| 3 | 1% SAP | 99 | 1 | 200 | 35 |
| 4 | 2% SAP | 98 | 2 | 200 | 35 |

To warrant consistent distribution of SAPs in the cementitious matrix, the PVA fibers and SAPs were first dry mixed with the cement for about 1 min. The sand was then included, and dry mixing progressed for another 1 min. Subsequently, water was added, and a superplasticizer (MasterGlenium 7700—BASF) was used to adjust workability at a constant flow for all mixtures. The amount of water was kept constant in all mixtures to compare the effect of each SAP dosage on the pore structure, crack self-healing, tensile and compressive strength of the tested mortars. The flow was determined using Standard Test Method for Flow of Hydraulic Cement Mortar (ASTM C1437). The flow in all mortar mixtures was 110 ± 5%. All the ingredients were then mixed for 2 min using a Hobart N50 mixer (Hobart Corporation, OH, USA). Cylindrical specimens (50-mm in diameter and 100-mm in height) were cast to determine the compressive and splitting tensile strength of the tested mortars. Cylindrical specimens incorporating 1% SAPs with 50 mm in diameter and 25 mm in height were made for X-ray μCT scan examination. Specimens were cured for 28 d in an environmentally controlled room at a temperature of 21 ± 1 °C and relative humidity ≥ 95%. After 28 days, all specimens were removed from the curing room and cracked.

### 2.2. Crack Creation and Measuring

For the computed tomography scan samples, the crack was created using a screw jack (Figure 1). The crack width was controlled via a calibration ruler as per the method described by Roig-Flores et al. [14]. In addition, an optical microscope was used to measure the width of the surface part of cracks. Based on the crack width range, specimens were

divided into three groups. For each environmental exposure, three groups of specimens with three different values of crack width were tested. The first group (small crack) consisted of three specimens having crack widths of 50 μm to 150 μm. The second group (medium crack) consisted of three specimens with crack widths of 150 μm to 300 μm. Finally, the third group (large crack) included three specimens with crack widths of 300 μm to 500 μm.

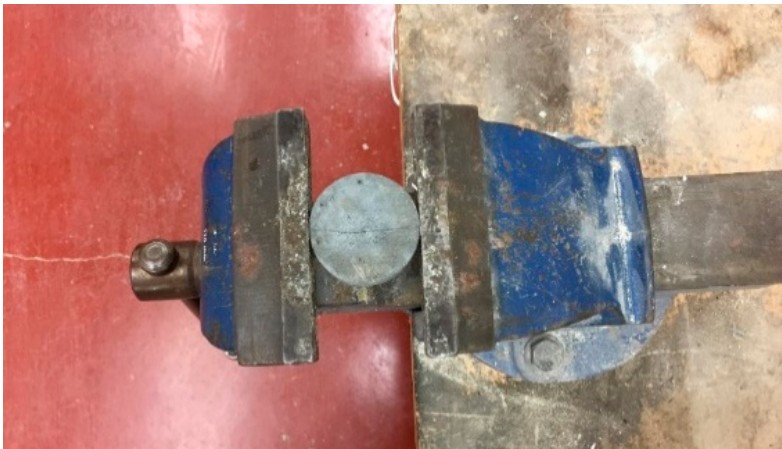

**Figure 1.** Cracking of specimens using screw jack.

### 2.3. Compressive and Tensile Strength

Three samples from each mixture were tested for compressive strength after 28 days of curing according to ASTM C39. In addition, another three (3) samples from each mixture were tested for tensile strength after 28 days of curing according to ASTM C496.

### 2.4. Environmental Exposure

The present study explores the crack width changes due to the self-healing of cementitious materials incorporating SAPs under a combined change in temperature and relative humidity. A set of samples from each group (small, medium, and large crack width) were subjected to different temperature and relative humidity cycles using a controlled climate chamber, as shown in Table 5. A similar set of samples were subjected to cyclic wetting and drying at 20 ± 2 °C, which consisted of storage in water for 4 d, followed by drying in the laboratory environment for 4 d. Finally, another set of samples (with similar crack width) was continuously submerged in water.

**Table 5.** Temperature and relative humidity cycles.

| Cycle # | Temperature (°C) | Relative Humidity % | Duration |
|:---:|:---:|:---:|:---:|
| 1 | 20 | 90 | 4 days |
| 2 | −10 | 20 | 4 days |
| 3 | 40 | 60 | 4 days |
| 4 | 20 | 90 | 4 days |

### 2.5. Pore Structure Analysis

The effect of SAPs on the mortar pore structure was examined using a Micromeritics AutoPore IV 9500 (Micromeritics Instrument Corporation, Norcross, GA, USA) mercury intrusion porosimeter. In addition, the effect of different environmental exposures on the pore structure of mortars was investigated. Before testing, pieces from previously cracked specimens were dried in a glass desiccator until a constant mass was reached.

### 2.6. Segmentation and Quantification Analysis of Crack

To measure the change in the entire volume of the crack due to self-healing, a Nikon XT H 225-ST industrial Computed Tomography (CT) scanning system was used. The XT H

225-ST provides detailed high-resolution inner and outer 3D images. The specimens were scanned before and after each environmental exposure. An advanced segmentation and quantification analysis software (Dragonfly 3.5) was used to calculate the change in the volume of crack due to self-healing.

### 2.7. Composition and Morphology of Healing Product

The composition and morphology of the healing products were investigated using a high-definition Hitachi SU3500 scanning electron microscope (SEM) imaging system coupled with energy dispersive X-ray (EDX) analysis. Before testing, all samples that exhibited self-healing were dried using a glass desiccator and then coated with gold.

### 3. Results and Discussion

#### 3.1. Compressive and Tensile Strength

Figure 2 depicts the compressive and tensile strength results of mortar specimens after 28 days. It can be observed that 0.5% SAP specimens achieved an increase in compressive and tensile strength. This can be ascribed to SAP particles which absorb some mixing water, leading to lower water-to-cement (w/c) ratio, and thus resulting in densification of the cementitious matrix. However, by increasing the amount of SAP particles in the cementitious matrix beyond that threshold level, both compressive and tensile strengths decreased, which may be attributed to the formation of macro-pores when SAP particles release the absorbed water at a later stage and shrink and delaying the hydration process due to water absorption during mixing.

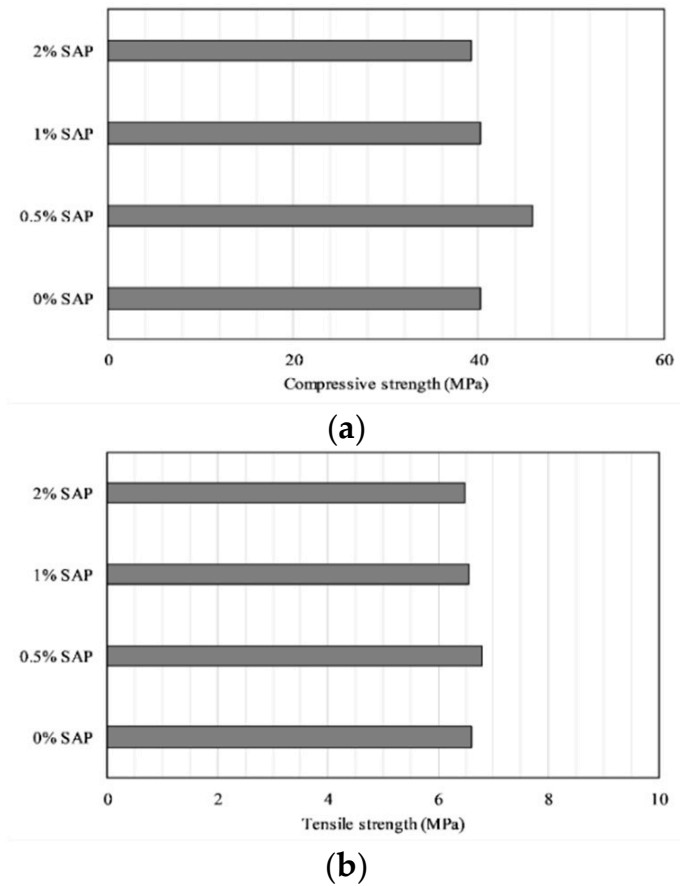

**Figure 2.** (**a**) Compressive, and (**b**) tensile strength of specimens (with/without SAP) after 28-d.

#### 3.2. Pore Structure

Figure 3a shows the MIP test results for cracked mortar specimens at 28 d (before self-healing). Results indicate that 0.5% SAP specimens had decreased porosity compared

with that of the control 0% SAP specimens since SAP particles absorb some mixing water, which reduces the w/c ratio and results in lower porosity. Similar findings were reported by Mönnig [32], who found that the total measured pore volume in specimens made with SAPs was smaller than that in specimens without SAPs. Snoeck et al. [33] also reported that cement pastes with SAPs and without additional water had a decrease in porosity compared to reference specimens with the same w/c ratio.

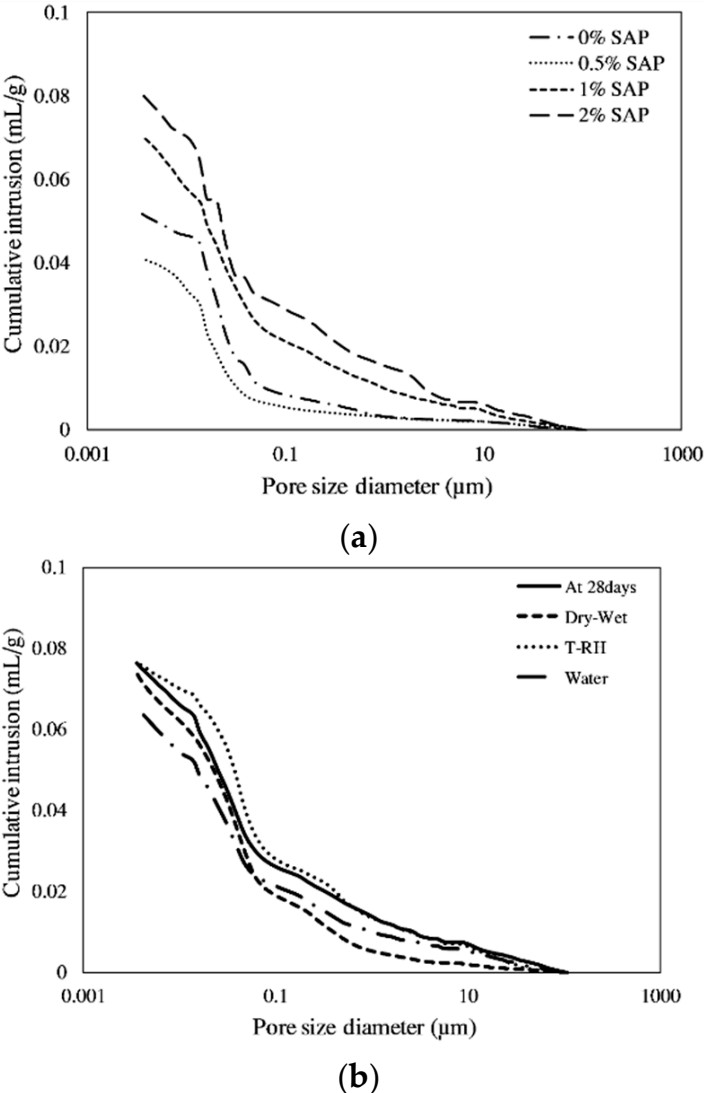

**Figure 3.** MIP results for cracked samples (**a**) at 28 d, and (**b**) samples with 1% SAP after five months of exposure to different environments.

On the other hand, MIP results show that the total porosity of cement mortars incorporating SAP increased as the SAP content increased beyond 0.5%. During hardening, SAP particles release their absorbed water and shrink, leaving behind voids in the cementitious matrix that have hundreds of microns in size [26,33]. For example, a previous study by Snoeck et al. [34] investigated the effect of high amounts of SAPs on the properties of cementitious materials. They posited that increasing the amount of SAP particles in cement mortars resulted in increasing the total amount of pores. This effect was more prominent with smaller SAP particles due to their higher surface area accessible for combining mixing water.

Figure 3b depicts the MIP test results for cracked mortar samples made with 1% SAPs and exposed to different environments. Results show that submerged samples and samples

subjected to cyclic wetting and drying achieved a reduction in porosity in comparison to that of samples exposed to an environment with a combined change in temperature and relative humidity. The presence of liquid water led to a decrease in porosity, which prompted microstructural densification. On the other hand, samples subjected to an environment with a combined change in temperature and relative humidity had a slight increase in porosity, which may be associated with the development of microcracks resulting from the cyclic exposure to different T and RH.

### 3.3. Surface Crack Healing
### 3.3.1. Cyclic T and RH

Figure 4 depicts the surface crack healing of mortar samples after five months of exposure to an environment with a combined change in temperature and relative humidity. It can be observed that only specimens with SAPs achieved partial healing. For example, specimens with 2% SAPs had crack closing of up to around 65 ± 10 µm. In comparison, 0% of SAP samples featured no crack closing via self-healing. For 1% and 0.5% SAP samples, the widest healed crack was 50 ± 10 µm and 20 ± 10 µm, respectively. Therefore, the capacity of crack healing in decreasing order was 2% SAP, 1% SAP, 0.5% SAP, and 0% SAP.

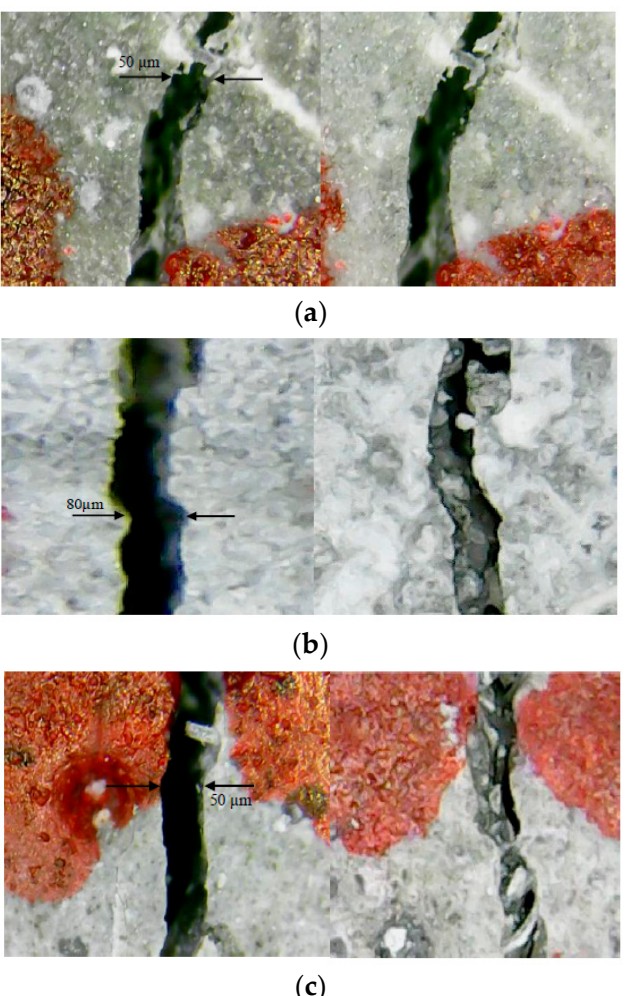

(**a**)

(**b**)

(**c**)

**Figure 4.** *Cont.*

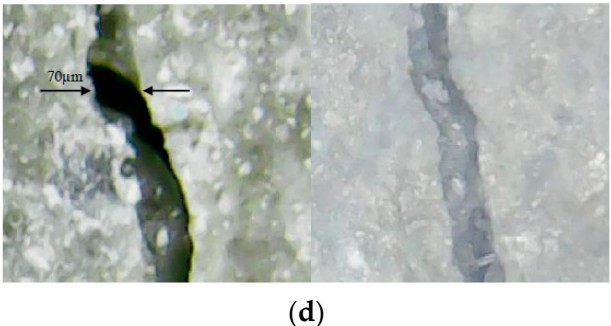

**(d)**

**Figure 4.** Surface cracks before and after five months of exposure to an environment with a combined change in T and RH of samples (**a**) 0% SAP; (**b**) 0.5% SAP; (**c**) 1% SAP; and (**d**) 2% SAP.

Results of EDX analyses (Figure 5) revealed that healing compounds formed in surface cracks were dominated by $CaCO_3$. Apparently, SAP particles were able to absorb moisture and stimulate autogenous healing, even without direct contact with water. Due to decreased RH, the initially swelled SAP particles could release water into the cementitious matrix. Previous study by Snoeck et al. [35] showed that only mortar specimens with SAPs exhibited self-healing due to moisture uptake when exposed to an environment with relative humidity higher than 60%. Similar specimens, yet with no SAPs, showed no sign of self-healing when exposed to a similar environment. According to Snoeck et al. [35], SAP particles are capable of sustaining cement hydration reactions through releasing their absorbed moisture and providing the necessary moisture for $CaCO_3$ to form and precipitate. Thus, SAPs could supply moisture to activate the autogenous self-healing in dry environments [25].

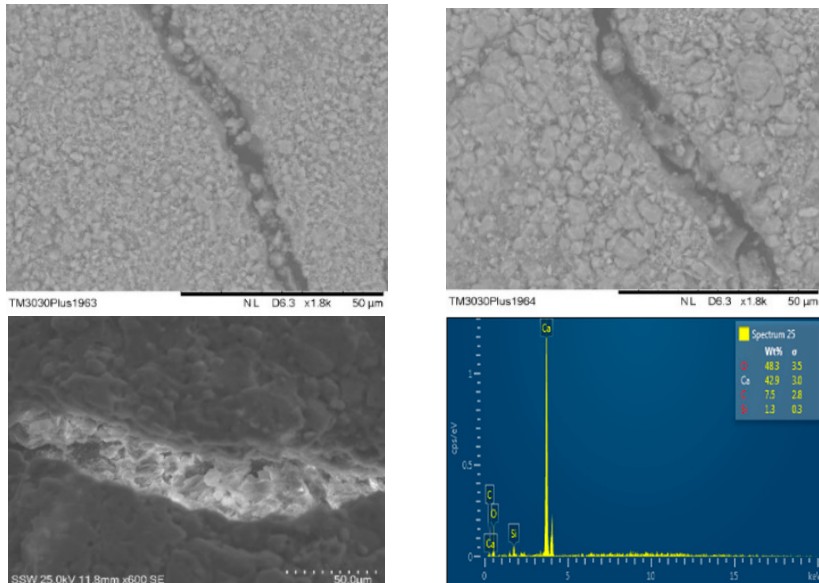

**Figure 5.** SEM image with EDX analysis of compounds within self-healed cracks of samples with 1% SAP after exposure to an environment with a combined change in T and RH.

### 3.3.2. Underwater Submersion

Figure 6 depicts the surface crack healing of fiber-reinforced mortar samples submerged in water for five months. For both types of samples, with and without SAPs, that were fully submerged into water, surface cracks having various widths exhibited self-healing. Healing products, which formed in cracks, consisted of white crystals. After five months of water submersion, only samples with initial crack widths of 50 to 150 μm (with and without SAPs) achieved complete surface crack healing.

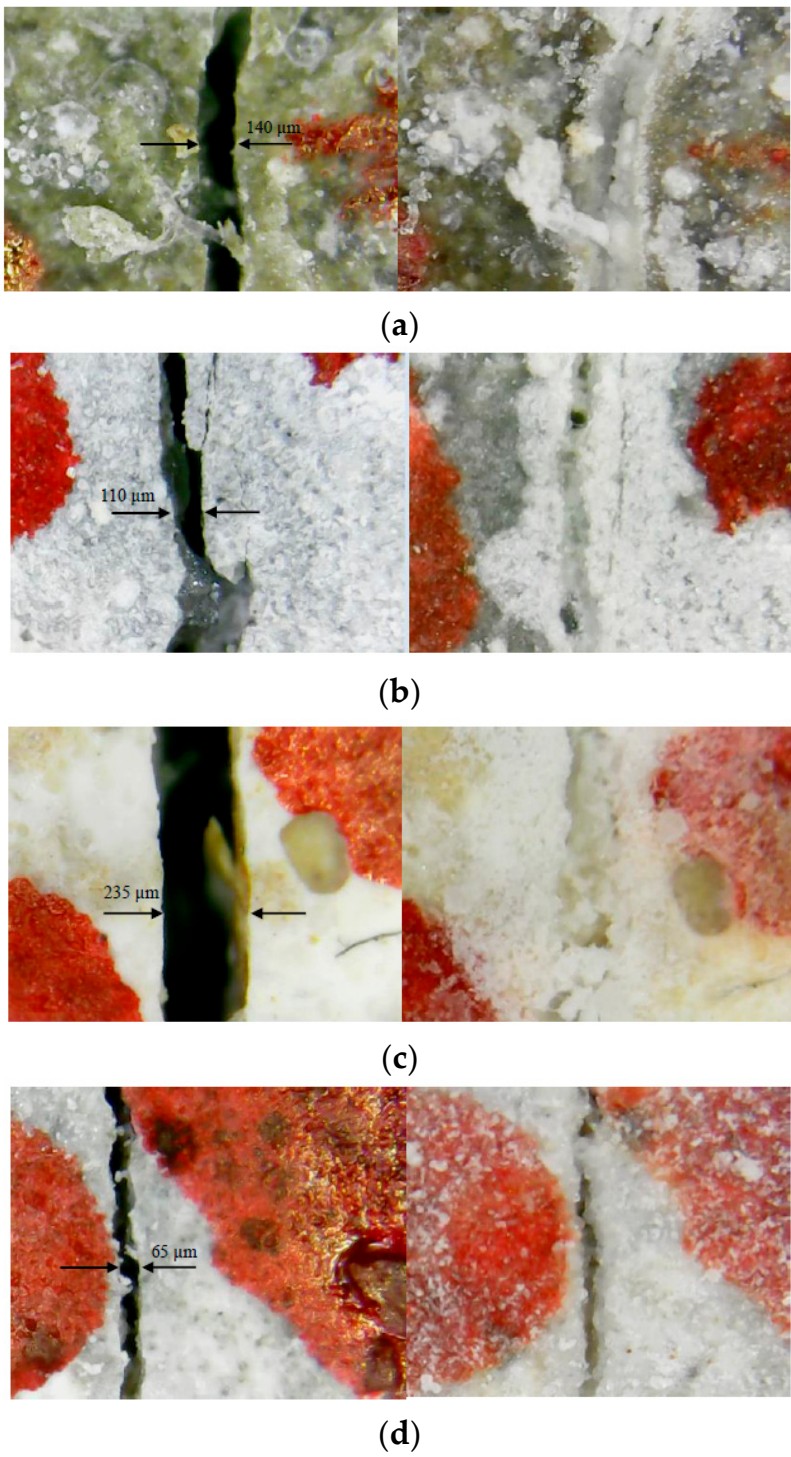

**Figure 6.** Surface cracks prior to and after five months of water immersion of samples. (**a**) 0% SAP; (**b**) 0.5% SAP; (**c**) 1% SAP; and (**d**) 2% SAP.

For the second group of specimens having initial crack width of 150–300 µm, surface cracks were only able to heal partially. The widest initial crack width that was healed for samples submerged in water for the five months test duration was about 275 ± 15 µm, 260 ± 10 µm, 255 ± 10 µm, and 235 ± 5 µm for the 0% SAP, 0.5% SAP, 1% SAP, and 2% SAP specimens, respectively.

For the third group of specimens having initial crack width of 300 to 500 µm, surface cracks achieved no or minor self-healing. Similar results were reported by others who found that the maximum surface crack width healed via autogenous or improved autogenous

self-healing, which uses crystalline, expansive, or supplementary cementitious materials, was less than 400 µm for specimens submerged in water [36–39]. According to Van Tittelboom et al. [36], as the crack width increased, crack self-healing became more difficult to achieve, indicating that crack self-healing is inversely proportional to the crack width. To characterize the self-healing products that developed within surface cracks, SEM with EDX detector was used. Results from SEM images and EDX spectrum of the healing compounds which formed within surface cracks (Figure 7) consisted primarily of oxygen, calcium, and carbon, indicating that it mostly consisted of calcium carbonate crystals.

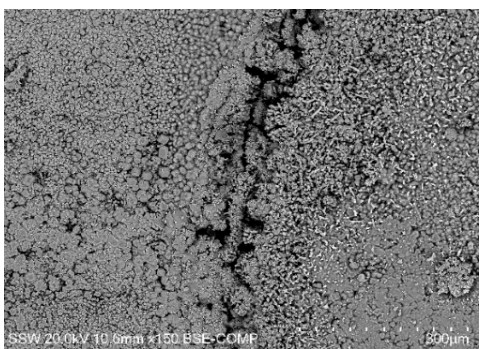 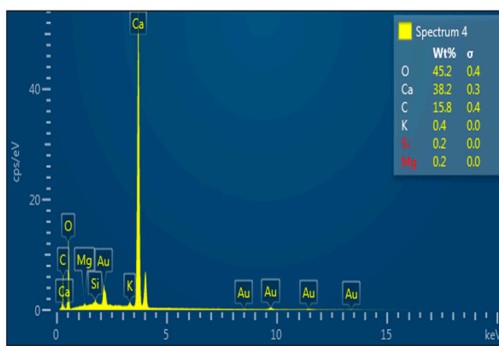

**Figure 7.** SEM micrograph with EDX pattern of self-healed compounds in cracks of samples incorporating 1% SAP after water submersion.

### 3.3.3. Cyclic Wetting and Drying

Figure 8 shows surface cracks before and after exposure to five months of cyclic wetting and drying. Results indicate that 2% SAP specimens achieved the highest self-healing rate. Surface cracks of up to around $220 \pm 10$ µm in width were completely sealed. On the other hand, 0% SAP samples achieved the least self-healing rate. The highest crack width healed in 0% SAP samples was $180 \pm 10$ µm. For 1% SAP and 0.5% SAP samples, the highest crack width healed was $205 \pm 5$ µm and $190 \pm 10$ µm, respectively. Therefore, the capacity of crack healing was in the order of 2% SAP, 1% SAP, 0.5% SAP, and 0% SAP, respectively.

These results can be ascribed to the fact that SAP particles were able to supply water during the dry cycle of the wet-dry process, therefore enhancing the potential of crack self-healing. A previous study by Snoeck et al. [24] investigated autogenous healing in strain-hardening cement-based composites containing SAPs. Crack healing was improved by adding SAPs in specimens exposed to wet/dry cycles. Therefore, SAPs can promote crack healing in cement-based materials exposed to cyclic wetting and drying. SEM and EDX analyses also showed that the healing products formed within surface cracks for samples exposed to wet/dry cycles consisted mainly of $CaCO_3$ (Figure 9). This finding agrees with previous studies [24,33,34]. Therefore, $CaCO_3$ appears to be the main autogenous self-healing compound of cracks in cementitious composites incorporating SAPs and exposed to cyclic wetting and drying.

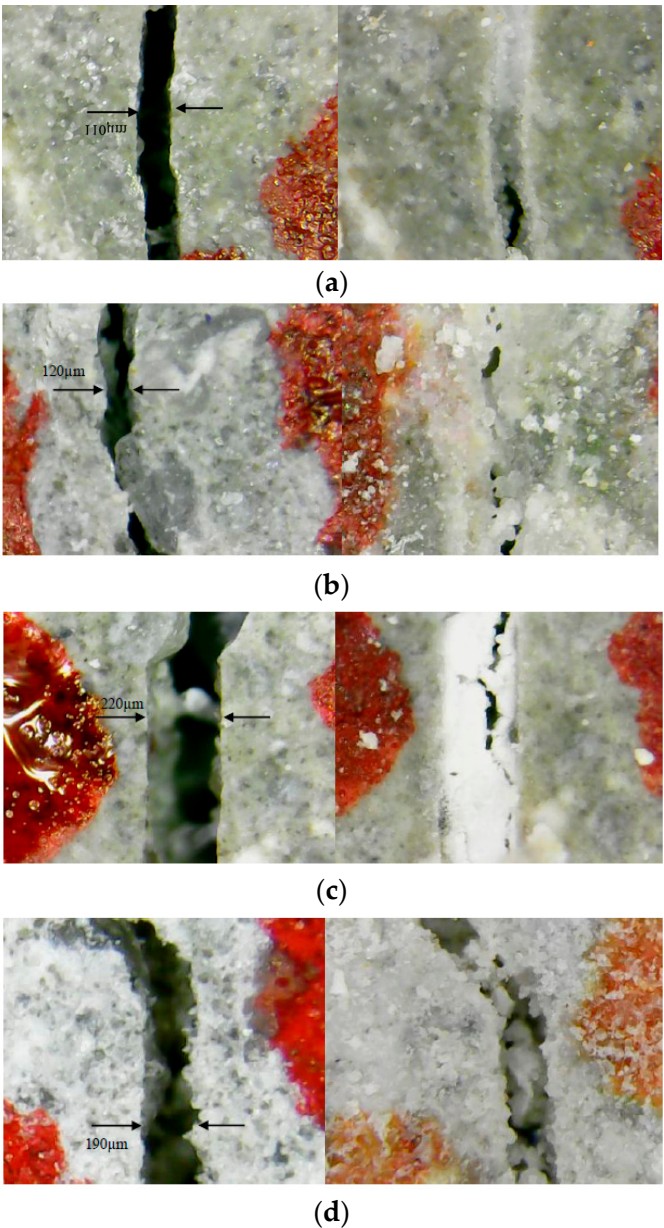

**Figure 8.** Surface cracks prior to and after five months of cyclic wetting and drying of samples (**a**) 0% SAP; (**b**) 0.5% SAP; (**c**) 1% SAP; and (**d**) 2% SAP.

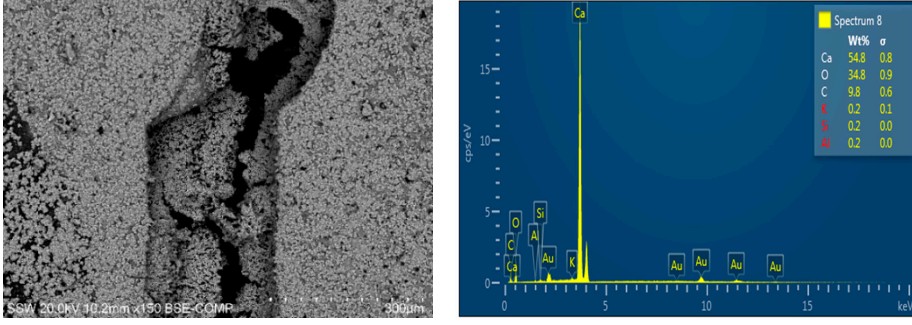

**Figure 9.** SEM image with EDX analysis of self-healed compounds within cracks of samples incorporating 1% SAP after cyclic wetting and drying.

### 3.4. Three-Dimensional Crack Quantification

To quantify the self-healing, a Nikon XT-H-225-ST μCT scanner housed at the Sustainable Archaeology lab, Western University, was used as shown in Figure 10. Samples made with 1% SAP having initial crack width of 150 to 300 μm and exposed to different environmental conditions were selected for CT scan analysis. Two scans were conducted (at the cracking and after exhibiting surface crack self-healing) for each sample. Parameters of the scan were defined as follows: voltage = 215 kVp; current = 80 μA (power of 17.2 W). A 1-mm Cu filter was used to filter the X-ray beam. A total of 3141 projections were gathered during a 56 min scan. The X-ray projections were harvested to reconstruct the exterior and interior of the sample using the Nikon's CT-Pro (v 4.4.3) reconstruction software. All CT scans were then analyzed using a 3D image processing and segmentation software (Dragonfly 3.5) to distinguish between the voids and cracks and determine the change in size of the entire crack volume owing to self-healing. For each sample, the entire crack was segmented, and the total volume was quantified before and after self-healing. The amount of the healing product was estimated by subtracting the volume of the crack before and after self-healing.

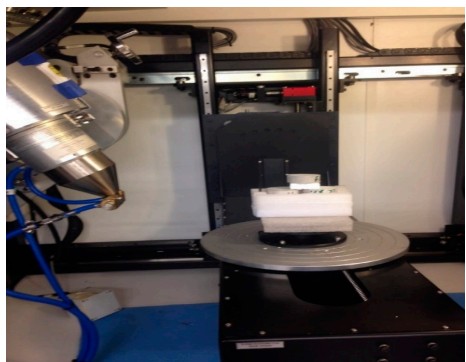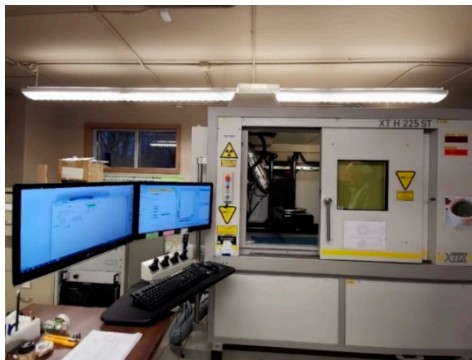

**Figure 10.** X-ray μCT scanning.

Figures 11 and 12 show the crack segmentation. Table 6 presents the results of quantification analysis of cracks using 3-D image analysis prior to and after each exposure. Results show that the healing efficacy was 26.32%, 19.11%, and 6.58% for 1% SAP samples exposed to water submersion, cyclic wetting, and drying, and an environment with a combined change in T and RH, respectively. The results indicate that the environmental condition plays a major role in autogenous crack self-healing of cement-based materials. For instance, according to the segmentation and quantification analysis, cement mortars incorporating 1% SAP and submerged in water achieved the highest crack volume reduction owing to autogenous self-healing in comparison to similar mortars that were exposed to either wetting and drying or to an environment with a combined change in T and RH.

Although mortar specimens incorporating SAPs and exposed to an environment with a combined change in T and RH exhibited patrial self-healing, CT scan analysis showed that their ability to self-heal was much less than that of their counterparts exposed to direct contact with liquid water. This indicates that cracks with direct contact with liquid water have a higher potential of autogenous self-healing. Similar findings were reported by Snoeck et al. [24] who investigated the autogenous self-healing behavior of strain-hardening cementitious materials containing SAP particles. They found that the autogenous self-healing was higher in specimens exposed to wet/dry cycles than in those exposed to only 90% relative humidity. Mousavi et al. [40] reported that SAPs can substantially increase the autogenous healing performance of concrete when exposed to wetting and drying cycles. According to Sisomphon et al. [38], cracks in cement-based materials exhibit autogenous self-healing when exposed to water due to $CaCO_3$ precipitation. For cracks to incur autogenous self-healing, both $H_2O$ and $CO_2$ need to be present in the crack. As the cracked samples start to release $Ca^{2+}$ ions, ($CaCO_3$) will precipitate and close the crack.

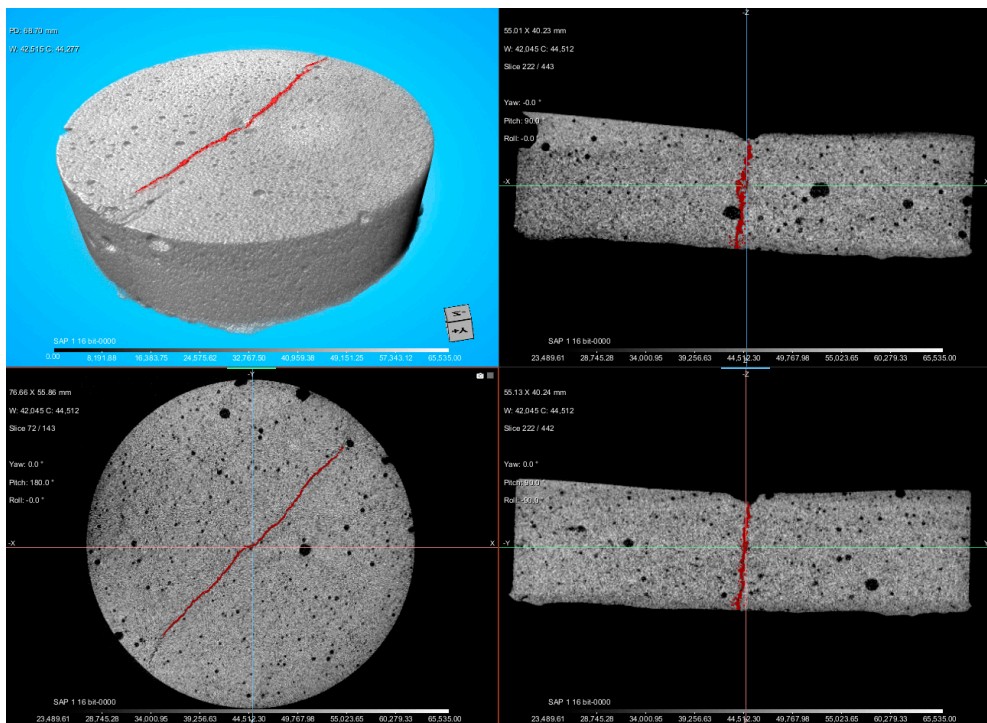

**Figure 11.** CT Scans images showing crack segmentation.

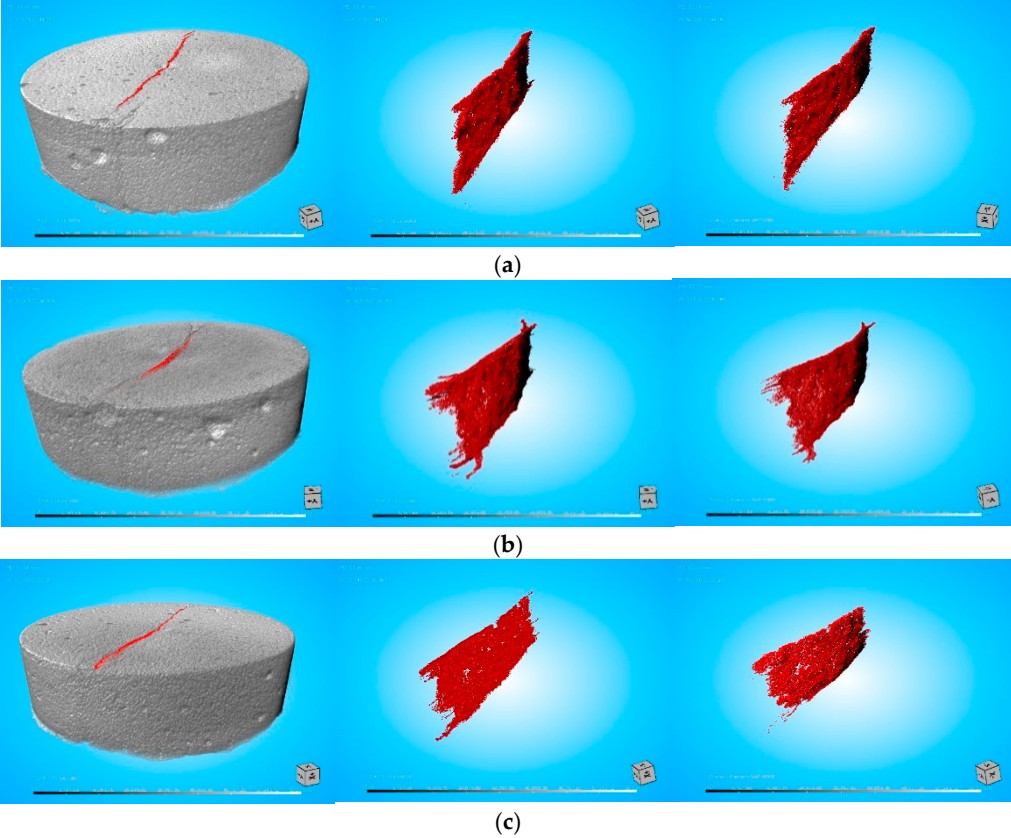

**Figure 12.** Crack segmentation before and after exposing to (**a**) an environment with a combined change in T and RH, (**b**) cyclic wetting and drying, and (**c**) continuous water submersion.

**Table 6.** Quantification of cracks prior to and upon self-healing.

| Environmental Exposure | Crack Volume ($\mu m^3$) | | Healing Efficiency (%) |
|---|---|---|---|
| | **before** | **after** | |
| Water submersion | $1.31 \times 10^{11}$ | $9.64 \times 10^{10}$ | 26.32 |
| Cyclic W/D | $1.89 \times 10^{11}$ | $1.53 \times 10^{11}$ | 19.11 |
| Cyclic T/RH | $1.68 \times 10^{11}$ | $1.57 \times 10^{11}$ | 6.581 |

CT scan images also showed that most healing compounds, which formed in specimens exposed to various environmental conditions, were essentially located close to the surface region of samples. Conversely, deeper at the interior regions of cracks, the amount of healing compounds was significantly less. Wang et al. [16] studied the self-healing behavior of concrete samples made with hydrogels encapsulated bacterial spores. Their results indicate that complete self-healing was only limited to surface cracks. Previous studies [41,42] explored the depth of autogenous self-healing in engineered cementitious composites using X-ray computed microtomography and found that self-healing was limited to the top surface of the crack until a depth of around 50 to 150 $\mu m$. Similar findings have been reported by Snoeck et al. [24]. A possible reason for this phenomenon is that the rapid formation of $CaCO_3$ at the surface cracks blocked water and carbon dioxide from entering deeper inside cracks, thus leading to less healing potential in the interior parts of cracks. According to Sisomphon et al. [38], locations near the crack surface region are usually the only areas that contain a significant amount of both calcium and carbonate ions. Therefore, more precipitation of $CaCO_3$ can be observed near the crack surface than in any other location.

## 4. Conclusions

In this study, the self-healing of cracks in cement-based materials incorporating SAPs under combined change of temperature and relative humidity was investigated and quantified using micro-computed tomography and three-dimensional image analysis. In addition, the influence of other environments such as continuous water submersion and cyclic wetting and drying was also studied and quantified. Based on the experimental results acquired in this study and pertinent discussion above, the following conclusions can be drawn:

1.  The segmentation and quantification analysis of X-ray $\mu$CT scans showed that the healing efficiency was 6.58%, 19.11%, and 26.32%, for samples made with 1% SAP and exposed to an environment with a combined change in T and RH, cyclic wetting and drying, and water submersion, respectively.
2.  The degree of self-healing was a function of the dosage of SAPs used and the type of environmental condition applied to the specimens.
3.  The effect of SAPs on crack self-healing was more pronounced in environments with less direct water contact.
4.  0.5% SAP specimens attained an increase in compressive and tensile strength. However, by increasing the amount of SAP particles in the cementitious matrix beyond that threshold level, both compressive and tensile strengths decreased.
5.  Surface cracks of up to around $65 \pm 10$ $\mu m$, $220 \pm 10$ $\mu m$, and $275 \pm 15$ $\mu m$ in width exhibited self-healing for specimens exposed to an environment with a combined change in T and RH, cyclic wetting and drying, and water submersion, respectively.
6.  The main crack self-healing compound formed in the fiber-reinforced mortar specimens with and without SAPs was $CaCO_3$.
7.  CT scan images revealed that self-healing of samples with and without SAPs was limited to the surface part of cracks.

**Author Contributions:** Conceptualization, M.L.N., A.R.S. and L.V.Z.; methodology, M.L.N., A.R.S. and L.V.Z.; software, A.R.S.; validation, A.R.S.; formal analysis, A.R.S.; investigation, A.R.S. and L.V.Z.; data curation, A.R.S. and L.V.Z.; writing—original draft preparation, A.R.S.; writing—review and editing, M.L.N.; visualization, A.R.S. and L.V.Z.; supervision, M.L.N.; project administration, M.L.N.; funding acquisition, M.L.N. All authors have read and agreed to the published version of the manuscript.

**Funding:** This research received no external funding.

**Institutional Review Board Statement:** Not applicable.

**Informed Consent Statement:** Not applicable.

**Data Availability Statement:** All data pertaining to the work conducted in this study is provided in the manuscript.

**Conflicts of Interest:** The authors declare that they have no conflict of interest.

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
