# Peer review of "Quantifying Crack Self-Healing in Concrete with Superabsorbent Polymers under Varying Temperature and Relative Humidity"

_sustainability, doi:10.3390/su132413999_

Round 1
Reviewer 1 Report
- Title of Table 2 should be revised to "Physical and chemical properties of sand".
- A detail explanation of the variation in material usage for each component in Table 4 is need.
- The units are needed in the ordinate of Figure 2.
- Since the specimen contains voids, please provide the method of distinguishing cracks and voids in CT scan images.
- The study presented a lot of experimental data, but most of its conclusions were qualitative. It is suggested to dig deeper into experimental data and get more instructive conclusions.
Author Response
The authors would like to thank the reviewer for the positive feedback, constructive criticism and insightful comments, which significantly improved our manuscript. We have carefully addressed each comment as explained in the attachment. All changes in the manuscript have been track changed in red color so that they can be easily retrieved.

Reviewer 2 Report
This study intends to experimentally determine the crack width variations due to the self-healing of SAP-contained mortars under a combined change of temperature and relative humidity. Results are interesting, as confirmed most of the results reported by the literature. The reviewer believes that major revision is necessary to add some explanations regarding SAP material and cracking of specimens:
- Abstract: please show some quantitative results at the end of the abstract.
- Abstract: please mention which dosages of SAP were used in this study.
- Table 3: what is the water absorption of SAP in the cement slurry. This parameter is important as additional water added to the mixture should be adjusted by this value.
- Table 4: there is additional water in mixtures due to SAP addition? Please mention slump (mini-slump for mortars) of mixtures if no additional water was added. Please also mention the content of the superplasticizer for each mixture.
- Section 2.1: please mention the production method and particle shape of SAP used.
- Table 4: literature frequently mentioned that 1.0% SAP is not good for mixtures, and we need to use low dosage. For instance, use the following reference to compare your results.
[ ] Mousavi, S. S., Ouellet-Plamondon, C. M., Guizani, L., Bhojaraju, C., & Brial, V. (2020). On mitigating rebar–concrete interface damages due to the pre-cracking phenomena using superabsorbent polymers. Construction and Building Materials, 253, 119181.
- Section 2.2: still, there’re is a question how did you control the crack? What was the rating load to produce cracks? How many specimens were considered for each group of cracked specimens?
- Fig. 2: the results are strange. Different conflicting results were also obtained by researchers for the compressive strength of SAP concrete. Please mention most of them in the text and compare your results. There is one concern regarding your mixture. If constant water and superplasticizer were used for mixtures, the energy used for compacting specimens was different, causing different samples. That is why more explanations are required. Having a constant slump for SAP samples is obligatory so that we can compare their results.
- Fig. 4 and 6: please add a scale bar.
Author Response

(The authors gave the same response as above.)

Round 2
Reviewer 2 Report
Dear Editor,
The manuscript contains interesting results. Although there is some concern regarding water absorption measurement of SAP, the reviewer thinks this paper can be published in your esteemed journal.
Best Regards,
Seyed Sina Mousavi